# Lung Function Tests, Quality of Life and Telemedicine: Three Windows on the Multifaceted World of Asthma in Adolescents

**DOI:** 10.3390/children9040476

**Published:** 2022-03-30

**Authors:** Eleonora Nucera, Angela Rizzi, Chiara Agrosì, Franziska Michaela Lohmeyer, Riccardo Inchingolo

**Affiliations:** 1UOSD Allergologia e Immunologia Clinica, Dipartimento Scienze Mediche e Chirurgiche, Fondazione Policlinico Universitario A. Gemelli IRCCS, 00168 Rome, Italy; eleonora.nucera@policlinicogemelli.it; 2Medicina e Chirurgia Traslazionale, Università Cattolica del Sacro Cuore, 00168 Rome, Italy; 3UOC Pneumologia, Dipartimento Scienze Mediche e Chirurgiche, Fondazione Policlinico Universitario A. Gemelli IRCCS, 00168 Rome, Italy; chiara.agrosi@guest.policlinicogemelli.it (C.A.); riccardo.inchingolo@policlinicogemelli.it (R.I.); 4Direzione Scientifica, Fondazione Policlinico Universitario A. Gemelli IRCCS, 00168 Rome, Italy; franziskamichaela.lohmeyer@policlinicogemelli.it

**Keywords:** asthma, adolescent, spirometry, impulse oscillometry, quality of life, parent–child relationship, caregiver, telemedicine, eHealth technology, COVID-19

## Abstract

Asthma is a heterogeneous disease usually characterized by chronic airway inflammation and recognized as the most prevalent chronic illness among children. Despite this, the knowledge as to how asthma affects adolescents is still scarce. One of the main management problems of asthmatic adolescents is the poor adherence to pharmacological and non-pharmacological treatments. The assessment of respiratory function and the impact on quality of life are still two crucial challenges in the management of asthmatic adolescents. Additionally, the COVID-19 pandemic has prompted physicians to explore complementary management strategies including telemedicine technologies. This review aims to provide an update on the contribution of respiratory functional tests, how asthma affects quality of life of adolescents and, finally, how telemedicine contributes to the management of adolescent asthmatics during the COVID-19 pandemic.

## 1. Introduction

Asthma is a heterogeneous disease that is usually characterized by chronic airway inflammation. It is defined by the history of respiratory symptoms such as wheeze, shortness of breath, chest tightness and cough that vary over time and in intensity, together with variable expiratory airflow limitation [1]. This chronic respiratory disorder is recognized as a major non-communicable disease in the pediatric population. According to the WHO (2013), adolescence can be seen as “the period in human growth and development that occurs after childhood and before adulthood, from ages 10–19” [2]. The 2021 update of the Global Strategy for Asthma Management and Prevention highlights the impact of rapid physical, cognitive and social changes on the management of asthmatic adolescents [1]. While persons of all ages suffer from asthma, certain age groups and their accompanying developmental stage challenge the implications of the disease even more. As a result, asthmatic adolescents may also have low treatment adherence toward preventive strategies and therefore high morbidity [3].

Despite the growing establishment of a personalized medicine approach [4] and modern omics methods [5] in diagnosis and management of asthma, the assessment of respiratory function [1] and the impact on quality of life [6] (QoL) in a "delicate" phase of growth are still two crucial challenges in asthmatic adolescents.

In addition, the COVID-19 pandemic has prompted clinicians to expand and, at times, change their follow-up tools for asthmatic patients, exploring the contribution of telemedicine to manage asthmatic teenagers [7].

The aim of this review is to provide an update from the last decade on (1) the role of respiratory functional tests in the diagnosis and management of asthmatic adolescents, (2) the impact of this chronic respiratory disease on adolescents’ quality of life and, finally, (3) the contribution of telemedicine to the management of asthmatic teenagers during the COVID-19 pandemic.

## 2. How to Explore Asthma in Adolescents: The Role of Lung Function Tests

Currently, the evaluation of respiratory function is primarily based on techniques such as spirometry, which includes forced vital capacity (FVC) and vital capacity (VC) maneuvers [8].

These spirometric maneuvers are used to diagnose, to predict the course of asthma and to estimate the risk of exacerbations [1]. In particular, the aforementioned spirometric investigations aim to characterize airways in terms of the presence and severity of airflow obstruction [8].

Asymptomatic asthmatic children may hide significant bronchial obstruction. Children with chronic airway obstruction may experience less breathlessness than children with acute obstruction [9]. In addition, poor perception of bronchial obstruction effects may place asthmatic adolescents in front of a greater risk of respiratory function decline and acute exacerbations [10]. Therefore, periodic evaluation of respiratory function is necessary to optimize the management of asthmatic children.

### 2.1. Flow-Volume Curves

The authors of the National Asthma Education and Prevention Program (NAEPP) guidelines recommend spirometry every 1–2 years for asthmatic children over 5 years of age [11].

The basics of spirometry apply to both pediatrics and adults, using the same principles for testing and equipment. Maximal expiratory flow-volume curves are considered the gold standard to assess respiratory function in asthmatic children [1]. In particular, periodic evaluation of pre- and post-bronchodilator forced expiratory volume in 1 second (FEV1) could help to identify children at risk of progressive respiratory function decline [12,13].

Furthermore, FEV1, in addition to being considered important to assess asthma severity [11,14], is an independent predictor of asthma exacerbations [15,16,17]; 1-year risk of asthma exacerbations is double in asthmatic children with a predicted baseline FEV1 <60% compared with children with normal values [17].

Recently, Hopp et al. updated data on lung function and bronchodilator response in asthmatic children, providing a plausible hierarchy of use of spirometric test results [18]. In particular, a 12% increase in FEV1 after bronchodilator, although considered an appropriate cut-off [19], could be potentially excessive in younger children. In fact, there is evidence that a threshold of more than 8% performed better than 12% in a study enrolling 1041 asthmatic children (mean age, 8.9 ± 2.1 years) from the Childhood Asthma Management Program [20]. More recently, Sottile et al. [21], studying 1146 white children aged 5 to 13 years from the ongoing CHildhood ASthma and Environment Research (CHASER) study (ClinicalTrials.gov ID: NCT02433275), showed that the probability of having asthma was almost nil when post-bronchodilator FEV1 change was less than or equal to 7.9%.

Although the pediatric population, the focus of this review, is expected to meet the American Thoracic Society/European Respiratory Society recommended end of forced expiration (EOFE) criteria [8], younger children could complete forced expiration in only 2 or 3 s. Despite this, FVC change after pharmacological bronchodilation test and a reduced total exhalation time after bronchodilation could help to investigate the presence of air trapping. In fact, Sorkness et al. identified an air-trapping obstruction phenotype among asthmatic children aged 6 to 17 years adopting a FVC z score of less than −1.64 or a FVC increase of at least 10% after bronchodilator [22].

However, FEV1 as well as the FEV1/FVC ratio are poorly correlated with the severity of symptoms; hence, asthma management should not be based solely on reported symptoms [23].

Historically, forced expiratory flow between 25 and 75% of FVC (FEF25-75) has been used to evaluate flow from the small airways. More precisely, FEF25-75 should be considered a measurement of flow at lower lung volumes. As in adults, the variability of the FEF25-75 is greater than that of the FEV1 and FVC [24]. Nevertheless, some studies suggested that FEF25-75 could be a clue of the symptomatic state in children [25,26,27]. In 2010, Simon et al. [28] found that the sensitivity of FEF25-75 to 65% of predicted value, in identifying a 20% change in FEV1 after bronchodilator in children enrolled in PACT [29] and CLIC [30] trials, was 90%.

### 2.2. Impulse Oscillometry

The forced oscillation technique (FOT) was developed in 1956 to explore respiratory functions [31]. A modification of this method is impulse oscillometry (IOS) [32]. In the latter, a loudspeaker delivers a regular, square pressure wave at a constant frequency to the respiratory system using spectral analysis from which all other individual frequencies are derived.

Compared to spirometry, IOS is a much simpler technique for assessing airway impedance and reactivity. IOS is effortless, is performed in tidal breathing and does not require special cooperation from the patient. In addition, IOS makes it possible to distinguish central versus peripheral airway obstruction [33,34].

The main indicators of IOS are: (1) resistance at 5 Hz (R5), (2) resistance at 20 Hz (R20), (3) the difference between R5 and R20 (R5−R20), (4) reactance at 5 Hz (X5), (5) reactance area (AX) and (6) resonant frequency of reactance (Fres). Because low oscillation frequencies can transmit more distally in the lungs than high frequencies, R5 reflects the respiratory system resistance, while R20 reflects the central airways resistance. R5−20 is an index of the peripheral airways. X5 represents the peripheral elastic resistance. AX reflects the elastic properties of the lung. As with low-frequency reactance, it provides important information about small airway obstruction. Finally, Fres is the oscillation frequency when the reactance is zero.

The usefulness of IOS has been examined in different respiratory diseases, including asthma [35]. The greatest advantage is the ability to monitor the course of the disease [36] and to assess therapy response [37,38], which is most notable in children because patient cooperation is only minimally required [39] and measurements are reproducable [40].

One of the main advantages of IOS, compared to other lung function tests, is that measurements of respiratory mechanics are as easy to take in preschool children as in schoolchildren and adolescents. In fact, in 2006, Tomalak et al. [41] demonstrated that all oscillometric resistances (at 5, 20 and 35 Hz) correlated significantly with plethysmographic airway resistance (Raw) in a group of 334 children aged 5–18 years. Furthermore, the strongest correlation was seen for R5 (r = 0.64) [41].

The strong relationship between oscillometric and pletismographic measurements allows to compare different populations and changes in respiratory function over the years in both physiological and pathophysiological conditions. In fact, in 2008, Nowowiejska et al. [42] studied 626 healthy polish children aged 3.1–18.9 years (278 boys and 348 girls) in order to define equations describing normal values of oscillatory parameters. The authors found that body height was the best predictor and resistance was best described with an exponential model, while reactances with a linear model, with correlation coefficient r reaching the value of 0.9. Furthermore, oscillometric resistances decreased with height, while reactances increased [42].

### 2.3. Exhaled Nitric Oxide

Measurement of exhaled nitric oxide (FeNO) through the mouth nitric oxide is a technique now used in children [43]. In the pediatric population, 20 parts per billion (ppb) is the cut point for low FeNO value, while an elevated FeNO value is above 35 ppb; finally, values between 20 and 35 ppb are considered indeterminant [44]. This procedure is an attractive additional tool in children because it does not involve any forced mechanics and is relatively simple to accomplish. The clinical use of FeNO in the pediatric population is still under investigation [45,46,47]. Recently, Lo et al. investigated the prevalence of abnormal FeNO values in 612 children aged 5–16 years with an existing asthma diagnosis or receiving asthma drugs [48]. A total of 36% of patients showed FeNO ≥35 ppb and 41.8% reported poor control [48]. More recently, Lo et al. demonstrated that high FeNO levels predicted asthma exacerbations during follow-up in a cohort of 460 children aged 5–16 years [49]. To date, there is evidence to conditionally recommend the use of FeNO as an additional tool for evaluating the treatment of asthma patients [44].

Limiting the contribution of respiratory function tests for diagnosing and monitoring adolescent asthma is the current COVID-19 pandemic context. Nevertheless, the possibility of carrying out oscillometric measurements and FeNO tests, with smaller devices compared to body plethysmographs, with the operator placed behind the child during breathing maneuvers, and with reduced acquisition times, could cushion the negative impact of new pandemic rises on the daily practice of pulmonary function laboratories [50]. 

## 3. Impact of Asthma on Quality of Life in Adolescents

Adolescence represents a critical developmental period as the individual loses the characteristics of childhood to fast reach psychic maturity [51]. Furthermore, the process of acceptance and internalization is much more complex than any other moment in life, making the adolescent more vulnerable to stress, which has a strong negative impact on the management of chronic diseases such as bronchial asthma [52]. As early as 2007, de Benedictis and Bush recognized that adolescents with asthma have specific needs and problems and should take separate attention [53]. In fact, while young children are completely dependent on their parents regarding their asthma care [54], and adults are capable of managing their own chronic illness effectively, adolescents find themselves somewhere in between [55].

These problems can include low levels of disease knowledge [56], denial of symptoms, non-adherence to asthma medications [57,58], unfruitful preventive asthma medication use [59], poor relationship between adolescents and their families [53] and a higher risk of morbidity, mortality and severe exacerbations [60].

Furthermore, growing evidence highlights the role of bronchial asthma in school absenteeism [61,62,63], its detrimental effect on at-school productivity (presenteeism) [63,64,65] and friend relations [66,67]. 

A negative effect on QoL is the inevitable and foreseeable consequence of these problems [68,69].

Growing scientific literature on the importance of QoL in pathologies has unveiled heterogeneity of the conceptualization, definition and operationalization [70]. This influences the interpretation of the increasing quality of life studies, even for asthmatic patients [71].

Table 1 shows studies published over the past 10 years exploring the QoL of asthmatic adolescents from 10 to 19 years of age.

Feinstein already observed over 30 years that QoL is often used as a generic term [75]. Exploratory tools such as dedicated questionnaires frequently replace a lacking explicit definition of QoL. As noted by Costa et al., these "patient-reported outcomes" are equally used as an umbrella term as they specify the data source, the patient, but do not explicitly specify outcomes [71].

The impact of asthma on individuals varies greatly from person to person according to their unique circumstances. According to Falvo, the impact is dependent on the individuals’ pre-illness personality, severity of the illness, the meaning the individual attaches to the illness, current living circumstances and the availability of family and social support [76]. Additionally, asthma not only significantly affects patients’ lives but also that of their friends, family, colleagues and the community [54].

Recent studies highlighted that the mother–adolescent relationship tends to be stronger than the father–adolescent relationship. This is probably due to the fact that woman generally take on most of the responsibility for child rearing and are more closely involved in their child’s life than men. Mothers who are more engaged in managing their children’s asthma complain more often of anxiety and depression than mothers of adolescents suffering from other chronic diseases [77,78,79,80].

Adolescents’ stressors are primarily centered on their asthma symptoms [81], and thus they affect their emotional well-being [82]. This in turn also raises stress levels of the family and affects their well-being negatively [73,74], which was described by Egan [82] as interactive stressors.

The family can be seen as a social network that gives family members their identity and provides strong psychological bonds. In addition to general support, families, especially parents, of children and adolescents with a chronic disease often fulfil the important role of the caregiver [83]. They provide physical, emotional and financial support. They rarely have any training, act often without recognition or support and hardly ever receive financial aid. Individuals living with chronic illness are often solely dependent on family caregivers [84].

Furthermore, the parent–child relationship changes during adolescence. This can be attributed not only to the generation gap as mentioned above but also to less time spent together and increased conflicts between family members and the adolescent. Parents often experience the period of adolescence as more challenging than other stages in their child’s life [85].

Finally, the actual COVID-19 pandemic begins to heavily influence QoL in asthmatic adolescents. In fact, Cekic et al. showed that asthmatic adolescents’ QoL, evaluated through EUROHIS-QOL 8 [86], a part of the World Health Organization Quality of Life (WHOQOL) questionnaire, is lower than in the healthy population [72].

## 4. eHealth in the Management of Asthmatic Adolescents

### 4.1. Definition

According to the WHO, eHealth is defined as the use of information and communication technologies (ICT) for healthcare services, such as public health control, research, education, disclosure and patient care [87]. eHealth is a large container which includes various methods. The first major distinction is between synchronous and asynchronous technologies. Synchronous technologies allow physicians to keep direct contact with the patient remotely without a face-to-face visit. These tools are effective in emergencies and can be used to promote multidisciplinary work allowing cooperation among group members, even at larger distances. Asynchronous technologies, where communication does not take place in real time, is more suited to routine situations and follow-up [88].

Main eHealth technologies are: mHealth (mobile Health), clinical intervention via mobile devices; telemedicine/telehealth, remote assistance via telephonic or electronic facilities; artificial intelligence (AI), remote assistance algorithm; robotic technologies; social networking applications, interactive web platforms; apps category; electronic health records (EHRs), digital version of patient’s paper chart [87,89,90].

In the COVID-19 era, eHealth technologies are progressively spreading because they keep virtual patients and physicians in contact, which reduces the necessity of face-to-face meetings [91]. This in turn resulted in decongested clinical services, and consequently interpersonal contacts were reduced, which led to reduced virus circulation. Especially for high-risk categories, hospital visits result to be safer [92]. There are also economic implications, such as saving Personal Protective Equipment (PPE), in a world of limited resources [93,94].

Furthermore, eHealth facilities, delivering remote services, would help guarantee adequate and quality care for patients living in rural areas despite disproportionate services distribution, reducing or eliminating the need for travel [95]. Additionally, multidisciplinary teams can exploit spreading virtual technologies to ensure cooperation among individual members of the team to promote global and accurate clinical assistance [96].

### 4.2. eHealth Technologies and Asthma

In the literature, several studies, focusing on asthma and telemedicine, include both adolescent and children, and little evidence exists that concentrates on adolescent patients.

A systematic review was carried out to investigate the experience of adolescents (15–24 years old) with mHealth in managing NCDs and identify strengths and weaknesses of these technologies (according to end users and implementers) [97]. Among the studies included in the review, one focused in particular on experiences of patients using an asthma self-management app for one week [98]; in another, an interactive system, using text-messaging (mASMAA), was developed for asthma management [99]. Users judged interventions as effective to monitor symptoms, to improve disease perception and to increase therapy adherence. However, some features were judged impractical (e.g., asthma management by recording peak flow) [98]. From the point of view of the implementers (health care professionals), the usefulness of the clinical appointment planning agenda was highlighted, allowing communication between patient and doctor and providing alerts on the worsening of clinical conditions [100,101]. Finally, the co-design between end users and implementers is important in developing eHealth technologies to make them more compliant to a patient’s everyday life [97].

Among diverse telemedicine applications, tele-rehabilitation is an emerging strategy to remotely carry out rehabilitation programs to improve the out-of-hospital management of chronic patients [102]. Dos Santos et al. performed a systematic review of randomized clinical trials on tele-rehabilitation in asthmatic children and adolescents (0 to 18 years old, 88.9% aged 6 years). In analyzed trials, heterogeneous innovative technologies (mainly the Internet, but also telephone (88.9%), video (33.3%) and audio records) have been compared with traditional rehabilitation control groups. Considering the wide age range, results might be influenced by constant active participation of parents required for infants [103].

Evidence shows that tele-rehabilitation is at least as effective as traditional care in selected groups of patients, such as the asthmatic juvenile population, with improvement of symptoms and QoL [104,105,106]. Regarding costs, evidence is instead contradictory [107,108]; considering the lack of evidence, it is impossible to say if tele-rehabilitation is cost-saving.

Due to the disproportionate distribution of health facilities, it is often difficult to find an asthma specialist in rural areas. Portnoy et al. conducted a study to investigate the possible equivalence between telemedicine and face-to-face visits for asthma management among children and adolescents (younger than 4 years, aged 4 to 11 years, and older than 12 years) [109]. In a 6-month follow-up period, all patients attended three clinical visits. In the telemedicine group, a digital stethoscope for auscultation, an otoscope and high-resolution camera to examine the throat were used during the real-time video call. A nurse was also responsible for instructing patients on the correct execution of spirometry maneuvers and the use of inhalers. Telemedicine proved to be not inferior to traditional visits resulting in high satisfaction rates.

In another systematic review and meta-analysis, eHealth technologies, especially mHealth, significantly increased adherence to inhaled corticosteroids in a generic population of asthmatic patients [87]. 

A randomized-controlled trial by Johnson et al., based on a short messaging service remind system, demonstrated an improvement in adherence to inhaled corticosteroid therapy in a small group of 12–17 year old patients [110]. At 3-weeks follow-up, the intervention group showed improvement (median change 4 to 6 days) in self-reported adherence to treatment (*p* = 0.011), in drug efficacy self-perception (*p* = 0.016) and in QoL (*p* = 0.037).

A recent systematic review analyzed effects of heterogeneous telemedicine forms in school age children [111]. Referring to adolescent age, this paper focused on two specific studies. The first one is a cohort study evaluating a group of asthmatic patients between 5 and 18 years old (mean age 13.4 ages ± SD 2.7) monitored in a 6-month follow-up based on periodic telemedicine visits. The authors reported significant improvements in patients’ QoL at week 4 and week 24 and caregivers’ QoL at week 24 [112]. In the second study, the effect of telepharmacy counseling on metered-dose inhaler technique and patient satisfaction were evaluated. Specifically, a pharmacist provided short (15 min) video consultations with each patient to explain and show the correct inhaler utilization. A scale from 0 to 8 was used to evaluate the patients’ technique as to how to use the inhaler. In the intervention group, the initial score was 3.80 and improved to 6.73 at 2–4 weeks follow-up. Instead, in the control group, the pretest value of 4.05 improved only to 4.86 with standard care (*p* < 0.001) [113].

Finally, numerous apps dedicated to asthma management exist and are in progressive development. For example, one of these apps was developed to help asthmatic children become more confident with their disease through a game in which they have to look after a special pet (an asthmatic fire-breathing dragon) [114].

New generations are digital natives, and this offers the opportunity to use communication technologies to offer healthcare assistance especially to adolescence affected by non-communicable diseases (NCDs), such as asthma [115]. For these patients, eHealth technologies could be used to improve disease awareness and coexistence with physical and mental negative effects [116]. 

Among the downsides of eHealth is to mention that privacy, sensitive data and adequate staff training (not just physicians) should be guaranteed at all times [113]. A possible limitation that should be better investigated in the future is whether eHealth technologies’ progressive diffusion could decrease the social gap because of improved delivery of care to all social classes and ethnicities or if eHealth increases this gap [114].

## 5. Future Prospective

Improved asthma control in adolescents is crucial to improve lung function, to optimize quality of life and to prevent potential adverse disease sequelae later in life.

Interventions should be made on a global level to specifically investigate adolescent patients.

In the future, scientific societies should harmonize multiple guidelines [1,117,118] about the use of objective tests to confirm/support the diagnosis of asthma.

To date, spirometry is strongly recommended as part of the diagnostic work-up of children aged 5–16 years with suspected asthma [119]. However, there is need for studies in adolescents assessing: the role/contribution of various available lung function tests (bronchodilator reversibility, peak flow variability, exhaled nitric oxide fraction, impulse oscillometry) and the ideal timing and the frequency of lung function tests to build the evidence base of adolescent asthma diagnosis.

Furthermore, future real-world studies should also focus on innovative strategies, appreciated by asthmatic adolescents, to promote medication adherence. Electronic monitoring devices, a new approach to achieving disease control and adherence in clinical practice and research [120], could be potentially suitable for this global aim.

Ultimately, a joint venture among patients, caregivers, doctors and researchers is crucial to find appropriate personalized strategies with positive effect on quality of life of patients and caregivers.

## Figures and Tables

**Table 1 children-09-00476-t001:** Studies focused on QoL in adolescent asthmatics (in the 10- to 19-year age bracket).

Author	Year	Country	Definition of QoL	Study Design	Number of Patients	Instrument
Dinglasan et al. [56]	2022	Malaysia	-	Cross-sectional	214	PAQLQ
Cekic et al. [72]	2022	Turkey	-	Cross-sectional	125	EUROHIS-QOL 8
Dut et al. [73]	2021	Turkey	-	Cross-sectional	122	PAQLQ BSI
Lang et al. [69]	2015	USA	-	Cross-sectional	58	PAQLQ, PACQLQ
Halterman et al. [74]	2011	USA	-	Prospective cohort	28	PAQLQ, PACQLQ

EUROHIS-QOL 8: 8-item measure for quality of life (QoL), derived from the WHOQOL-100 and the WHOQOL-BREF. The overall QoL score is formed by a simple summation of scores on the eight items, with higher scores indicating better QoL. PAQLQ: Pediatric Asthma Quality of Life Questionnaire. BSI: Brief symptom inventory. PACQLQ: Pediatric Asthma Caregiver’s Quality of Life Questionnaire.

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
