# Peer review of "Lung Function Tests, Quality of Life and Telemedicine: Three Windows on the Multifaceted World of Asthma in Adolescents"

_children, 2022, doi:10.3390/children9040476_

Round 1
Reviewer 1 Report
Please add recent work of Hopp et al as a reference for PFT.
Hopp RJ, Wilson, MC, Pasha MA. (2021): A compendium and review of pediatric pulmonary function testing assessment opportunities for asthma, Journal of Asthma, DOI: 10.1080/02770903.2021.1941094
IOS and FeNO could be done with less restrictions in the COVID era; and should be included and referenced. This would be critical if future COVID flares occur
Author Response
March, 5th 2022
To Guest Editor and Reviewers
Children MDPI
We would like to greatly thank the Guest Editor and Reviewers who encouraged a revision of the manuscript.
Please find the enclosed the Revision vers. 1 of the Review article entitled “Lung function tests, quality of life and telemedicine: three windows on the multifaceted world of asthma in adolescents” by Eleonora Nucera, Angela Rizzi, Chiara Agrosì, Franziska Michaela Lohmeyer and Riccardo Inchingolo.
[Children] Manuscript ID: diagnostics-1614044- Major Revisions
Author's Reply to the Review Report (Reviewer 1)
Comments and Suggestions for Authors
Please add recent work of Hopp et al as a reference for PFT.
Hopp RJ, Wilson, MC, Pasha MA. (2021): A compendium and review of pediatric pulmonary function testing assessment opportunities for asthma, Journal of Asthma, DOI: 10.1080/02770903.2021.1941094
We thank the Reviewer for the comment. We modified the manuscript according to Reviewer’s comments. Moreover, we extended PFT section adding data on FEF25-75 and FeNO.
IOS and FeNO could be done with less restrictions in the COVID era; and should be included and referenced. This would be critical if future COVID flares occur.
We thank the Reviewer for the comment. We modified the manuscript according to Reviewer’s comments.
With the best regards,
Eleonora Nucera, Angela Rizzi, Chiara Agrosì, Franziska Michaela Lohmeyer and Riccardo Inchingolo
Corresponding Author:
Riccardo Inchingolo, MD, PhD
UOC Pneumologia, Fondazione Policlinico Universitario A. Gemelli IRCCS. Largo A. Gemelli, 8 – 00168 – Rome, Italy.
riccardo.inchingolo@policlinicogemelli.it
Corresponding Author will receive all editorial communications
The authors declare that the manuscript, or specified parts of it, have not been and will not be submitted elsewhere for publication.
Reviewer 2 Report
Dear Authors,
I read the article with big interest. Before the publication i would like to ask you to consider a minor following comments:
The affiliation of the authors should be, in my opinion, in English language.
Abstract:
Authors wrote: “This review aims to provide an update on, which function assume
respiratory functional tests…” I don’t understand what authors mean about which function assume respiratory functional tests, please paraphrase the sentence.
2.1. Flow-volume curves.
Authors should describe more important parameters and write why they are useful or not useful in asthma assessment in children/adolescents. This section is too short, it seems like asthma diagnosis is very easy while it is not.
I don’t know why authors more focused on impulse oscillometry in comparison to spirometry.
In quality of life, section 3 I would write more specific informations about how asthma affects adolescents life. For example, I suggest to show the average absence from school classes because of asthma.
The section eHealth in the management of asthmatic adolescents is well written and interesting.
No need to add extra space between subparagraphs.
Kind regards
Author Response
March, 5th 2022
To Guest Editor and Reviewers
Children MDPI
We would like to greatly thank the Guest Editor and Reviewers who encouraged a revision of the manuscript.
Please find the enclosed the Revision vers. 1 of the Review article entitled “Lung function tests, quality of life and telemedicine: three windows on the multifaceted world of asthma in adolescents” by Eleonora Nucera, Angela Rizzi, Chiara Agrosì, Franziska Michaela Lohmeyer and Riccardo Inchingolo.
Author's Reply to the Review Report (Reviewer 2)
Comments and Suggestions for Authors
I read the article with big interest. Before the publication I would like to ask you to consider a minor following comments:
The affiliation of the authors should be, in my opinion, in English language.
We thank the Reviewer for the comment. In accordance with the indications of our Institution, we reported affiliations in Italian.
Abstract:
Authors wrote: “This review aims to provide an update on, which function assume respiratory functional tests…” I don’t understand what authors mean about which function assume respiratory functional tests, please paraphrase the sentence.
We thank the Reviewer for the comment. We modified the sentence explicating the focus on the contribution of respiratory function tests.
2.1. Flow-volume curves.
Authors should describe more important parameters and write why they are useful or not useful in asthma assessment in children/adolescents. This section is too short, it seems like asthma diagnosis is very easy while it is not.
I don’t know why authors more focused on impulse oscillometry in comparison to spirometry.
We thank the Reviewer for the comment. We extended “Flow-volume curves” section and described more important parameters (FEV1, FVC, FEF25-75 and their change after bronchodilator). Furthermore, we added a specific section focused on exhaled Nitric Oxide.
In quality of life, section 3 I would write more specific information about how asthma affects adolescents life. For example, I suggest to show the average absence from school classes because of asthma.
We thank the Reviewer for the comment. The section 3 was modified according Reviewer’s comments: we described the impact of asthma on school absenteeism, at-school productivity and friend relations.
The section eHealth in the management of asthmatic adolescents is well written and interesting.
We thank the Reviewer for the appreciation.
No need to add extra space between subparagraphs.
We thank the Reviewer for the comment. We modified the manuscript according to Reviewer’s comment.
With the best regards,
Eleonora Nucera, Angela Rizzi, Chiara Agrosì, Franziska Michaela Lohmeyer and Riccardo Inchingolo
Corresponding Author:
Riccardo Inchingolo, MD, PhD
UOC Pneumologia, Fondazione Policlinico Universitario A. Gemelli IRCCS. Largo A. Gemelli, 8 – 00168 – Rome, Italy.
riccardo.inchingolo@policlinicogemelli.it
Corresponding Author will receive all editorial communications
The authors declare that the manuscript, or specified parts of it, have not been and will not be submitted elsewhere for publication.
Round 2
Reviewer 1 Report
The manuscript is a series of statements without links.
The explanation of asthma is acceptable, but it's applicability of when patients are seen on telehealth is missing, and probably isn't relevant.
The explanation of pulmonary function is fine, but the link to telehealth and not being able to test is missing; however the timing of COVID restriction lifting makes this less relevant.
Also, what asthma test results SHOULD be used. Not clear at all
The explain of QOL in ASTHMA IS FINE, BUT WHAT WAS THE EFFECT OF COVID isolationism?
The explanation of Telehealth was fine. But tying it back to specific asthma telehealth interaction was missing
I will only accept if a major revision is submitted.
Author Response
March, 15th 2022
To Guest Editor and Reviewer
Children MDPI
We would like to greatly thank the Guest Editor who encouraged a revision of the manuscript.
[Children] Manuscript ID: diagnostics-1614044- Major Revisions
Author's Reply to the Review Report (Reviewer 1)
Comments and Suggestions for Authors
The manuscript is a series of statements without links.
We respect the Reviewer's comment, but we disagree with him/her. Our review focuses on three distinct but equally relevant aspects relating to adolescent asthma, as explained in the summary presented to the Guest Editor.
The explanation of asthma is acceptable, but it's applicability of when patients are seen on telehealth is missing, and probably isn't relevant.
We thank the Reviewer for the comment. Since the popularity of telemedicine during the pandemic, the results of many studies are currently in press and results are coming.
The literature documents a growing interest of the scientific community in telemedicine as a resource in the management of asthmatic patients, with over 180 papers published in the last 2 years with “asthma” and “telemedicine” as key words of the research on pubmed. We agree with the Reviewer on the need of prospective studies with statistical power to explore the benefits of this tool and to compare telemedicine vs in-person asthma care, in particular during this critical period of growth. We stressed this in Future Prospective section.
The explanation of pulmonary function is fine, but the link to telehealth and not being able to test is missing; however the timing of COVID restriction lifting makes this less relevant.
We respect the Reviewer's comment, but we disagree with him/her. As previously explained, our review focuses on three distinct but equally relevant aspects relating to adolescent asthma.
Also, what asthma test results SHOULD be used. Not clear at all.
We respect the Reviewer's comment, but we disagree with him/her. To date, there is no single lung function index or single set of lung function tests that allow a definitive diagnosis of asthma based solely on respiratory functional assessment. In our manuscript, we have broadly extended the section of lung function tests, welcoming comments from both Reviewers (1st round).
The explain of QOL in ASTHMA IS FINE, BUT WHAT WAS THE EFFECT OF COVID isolationism?
We respect the Reviewer's comment, but we disagree with him/her. The impact of COVID on quality of life is beyond the scope of our review as well as the call of Guest Editor.
The explanation of Telehealth was fine. But tying it back to specific asthma telehealth interaction was missing.
Sorry, but we don't understand this comment.
I will only accept if a major revision is submitted.
Sorry, but we don't understand the comment. In particular, the statement of the Reviewer is clearly in contrast with what he / she reported in the first round of review, in which he / she requested the insertion of a citation. Furthermore, it is curious that although most of the Reviewer’s comments in the second round are characterized by appreciation of various aspects of the manuscript, the Reviewer ultimately requests major revisions. Therefore, we leave the final choice to the Guest Editor.
With the best regards,
Eleonora Nucera, Angela Rizzi, Chiara Agrosì, Franziska Michaela Lohmeyer and Riccardo Inchingolo
Corresponding Author:
Riccardo Inchingolo, MD, PhD
UOC Pneumologia, Fondazione Policlinico Universitario A. Gemelli IRCCS. Largo A. Gemelli, 8 – 00168 – Rome, Italy.
riccardo.inchingolo@policlinicogemelli.it
Corresponding Author will receive all editorial communications
The authors declare that the manuscript, or specified parts of it, have not been and will not be submitted elsewhere for publication.